# Root Reinforcement in Slope Stability Models: A Review

**Elena Benedetta Masi** *[ID], **Samuele Segoni** [ID] **and Veronica Tofani**

Department of Earth Sciences, University of Florence, Via La Pira 4, 50121 Florence, Italy;
samuele.segoni@unifi.it (S.S.); veronica.tofani@unifi.it (V.T.)
* Correspondence: elenabenedetta.masi@unifi.it

**Abstract:** The influence of vegetation on mechanical and hydrological soil behavior represents a significant factor to be considered in shallow landslides modelling. Among the multiple effects exerted by vegetation, root reinforcement is widely recognized as one of the most relevant for slope stability. Lately, the literature has been greatly enriched by novel research on this phenomenon. To investigate which aspects have been most treated, which results have been obtained and which aspects require further attention, we reviewed papers published during the period of 2015–2020 dealing with root reinforcement. This paper—after introducing main effects of vegetation on slope stability, recalling studies of reference—provides a synthesis of the main contributions to the subtopics: (i) approaches for estimating root reinforcement distribution at a regional scale; (ii) new slope stability models, including root reinforcement and (iii) the influence of particular plant species, forest management, forest structure, wildfires and soil moisture gradient on root reinforcement. Including root reinforcement in slope stability analysis has resulted a topic receiving growing attention, particularly in Europe; in addition, research interests are also emerging in Asia. Despite recent advances, including root reinforcement into regional models still represents a research challenge, because of its high spatial and temporal variability: only a few applications are reported about areas of hundreds of square kilometers. The most promising and necessary future research directions include the study of soil moisture gradient and wildfire controls on the root strength, as these aspects have not been fully integrated into slope stability modelling.

**Keywords:** root reinforcement; slope stability; distributed model; shallow landslides; vegetation

## 1. Introduction

Vegetation plays a crucial role in protecting people, settlements and infrastructures from hydrogeomorphic hazards (e.g., [1,2]). It strongly affects mechanical and hydrological soil behavior, particularly related to shallow landslides, frequent mass movements widespread worldwide [3]. Shallow landslides constitute one of the most hazardous categories of mass movements, mainly because of their frequent evolution into rapid mass movements, assuming characteristics of debris avalanches and flows [4]. Indeed, commonly, this kind of landslide highly increases their velocity and involves increasing volumes of mobilized material as it propagates downstream [5]. Furthermore, these landslides are mainly triggered by intense rainfall, meaning that it is rare that only single failures happen, indeed, multiple and diffused landslide events are instead commonly triggered in the region hit by intense rainfall [6].

The vegetal communities have a stabilizing action in the vadose zone of the slopes [7,8], mainly due to (i) influence on soil suction by root water uptake, (ii) reinforcement of the soil due to the presence of roots, increasing the tensile strength, iii) anchoring of the shallowest layers to the deep and usually more stable substrates, (iv) surcharge due to weight of plant biomass (aerial part and root system) that increases the normal stresses to the slope and (v) rainfall interception by canopy and evapotranspiration, reducing the delivery rates of intense precipitation and lowering the water table.

On the other hand, some effects due to the presence on the slopes of the vegetation cover have a destabilizing action [9,10], such as (i) increasing the parallel stresses due to the plant weight, ii) transmission the bending moments by canopy through stems and roots and (iii) wedging the roots into rock fractures. Nonetheless, except for in particular contexts, it is largely recognized that the presence of plants constitutes a mitigating element for slope instabilities. In this sense, root reinforcement—$c_r$, the increase of the tensile strength of soils due to the root network, also called "root cohesion"—is the most relevant factor from the mechanical point of view (e.g., [11,12]).

Considering the influence of vegetation in slope stability analysis represents a significant and still open challenge for research. Relations established between plants and the surrounding environment are many, complex and constantly changing. Any effect of vegetation that can be treated in the slope stability studies poses a challenge for the research. The variables that determine the magnitude of the vegetation effects in space and time are many and their evaluation in broad areas represents a significant limit for distributed slope stability analysis; the treatment of the mechanical effect of root reinforcement is no exception to this problem.

Research within the framework of slope stability has increasingly pointed towards the analysis and quantification of the influence exerted by the vegetation on the mechanisms involved in the triggering of shallow landslides, and the literature on this argument has grown tremendously in recent years.

The present review aims to summarize the most recent studies about the vegetation effects in slope stability dynamics, focusing on the root reinforcement effect and its parameterization into slope stability models. The first part of the paper introduces the main effects of vegetation on slope stability (with a focus on the root reinforcement), recalling studies and authors of note of the last few decades. The second part is dedicated to most recent tests and modeling about vegetation and slope instabilities: first, the evaluation of root reinforcement in wide areas is analyzed with reference to the most recent studies; then, studies dealing with development of slope stability models that consider root reinforcement are reviewed, followed by works on the influence on slope stability of some plant species, forest management techniques, wildfires and moisture gradients.

## 2. Hydrological and Mechanical Effects of Vegetation on Slope Stability

The land cover of plant species affects soil behavior through many hydrological and mechanical processes (Figure 1) (e.g., [7,8,13]). At the catchment scale, the hydrological effects of interception, suction, evapotranspiration and infiltration strongly affect the soil water budget and runoff processes, while at the local scale, the mechanical effect of root reinforcement is the leading factor for slope stability [11].

The main hydrological effect of vegetation is delaying the soil from reaching the critical saturation level, which can trigger mass movements by means of several processes (e.g., [14]).

The processes causing this effect are:

- Suction: vegetation affects soil moisture by a root-water uptaking process driven by transpiration [15]: moisture is adsorbed by roots from the surrounding soil (trees can reduce soil moisture levels from distances up to three times the crown radius [12]), inducing soil suction (e.g., [16]); in regions where precipitation consistently exceeds the potential evapotranspiration, the soil moisture detraction by the latter is negligible; nevertheless, in case of moderate rainfall events, evapotranspiration may reduce soil moisture before the rainfall, increasing the amount of water storable in the soil [11];
- Interception: the effect of this process is the redistribution of gross rainfall falling onto plant surfaces, the rain is temporarily retained and successively lost in the atmosphere by evaporation or flows-drops onto the ground [17]; the magnitude of this phenomenon depends on climate, plant species and age, and forest structure (e.g., [18,19]); branches and foliage are capable of intercepting and then evaporating

nearly all rainfall in case of low intensities, while during high-intensity rainfall, plants can intercept only a fraction [20];

- Infiltration and subsurface flows: flow pipes and channels left by root decay can help slopes to drain faster [21,22]; on the other side, root channels can also increase infiltration rate, raising landslide hazard [11].

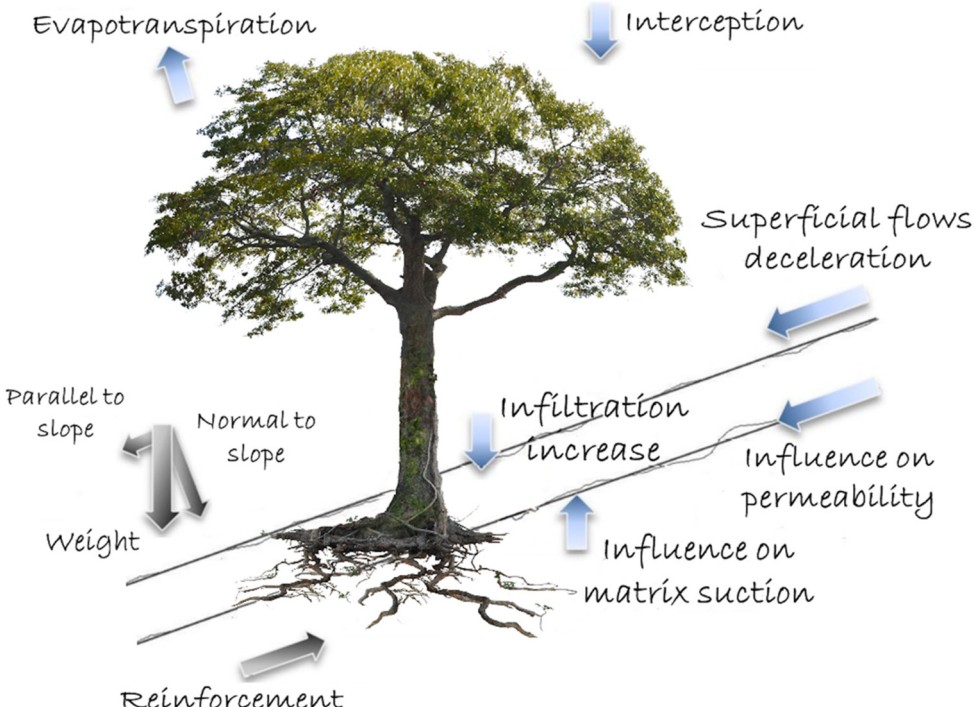

**Figure 1.** Mechanical and hydrological main effects of vegetation on slopes' soil behavior.

The mechanical behavior of soil can be affected by the following processes worked by vegetation:

- Soil reinforcement: root systems of plants increase the shear strength of soils through a combined action by the large and the small roots (Figure 2); large woody roots can anchor the superficial soil layers to more stable substrates crossing potential planes of weakness, while small roots strengthen bonds with the soil particles, increasing the overall cohesion of the soil–roots matrix; the reinforcement by roots can work on the basal failure plane of a landslide or on lateral failure (e.g., [23–25]);
- Surcharge: vegetation (particularly trees) weight increases both the normal and the tangential forces acting on slopes, but generally, the influence of this factor on the slope stability is negligible [26,27]; the surcharge due to the weight of mature forest of beech for instance is unlikely higher than 2.5 kPa, the equivalent of a layer of stony soil 15 cm thick [28,29];
- Buttressing and arching between plants: roots, stems and branches of trees can work as buttress piles or arch abutments constituting a resistance force against shear forces acting on the slopes [30,31];
- Deep anchoring: particularly in shallow soils of upland areas, roots can penetrate into bedrock joints and fractures attaining a strong and deep anchoring for the plant [32,33];
- Rock fracturing: plant roots can promote physical weathering processes of bedrocks by wedging and growing into pre-existing bedrock fractures or creating new fractures in combination with chemical processes [32,34].

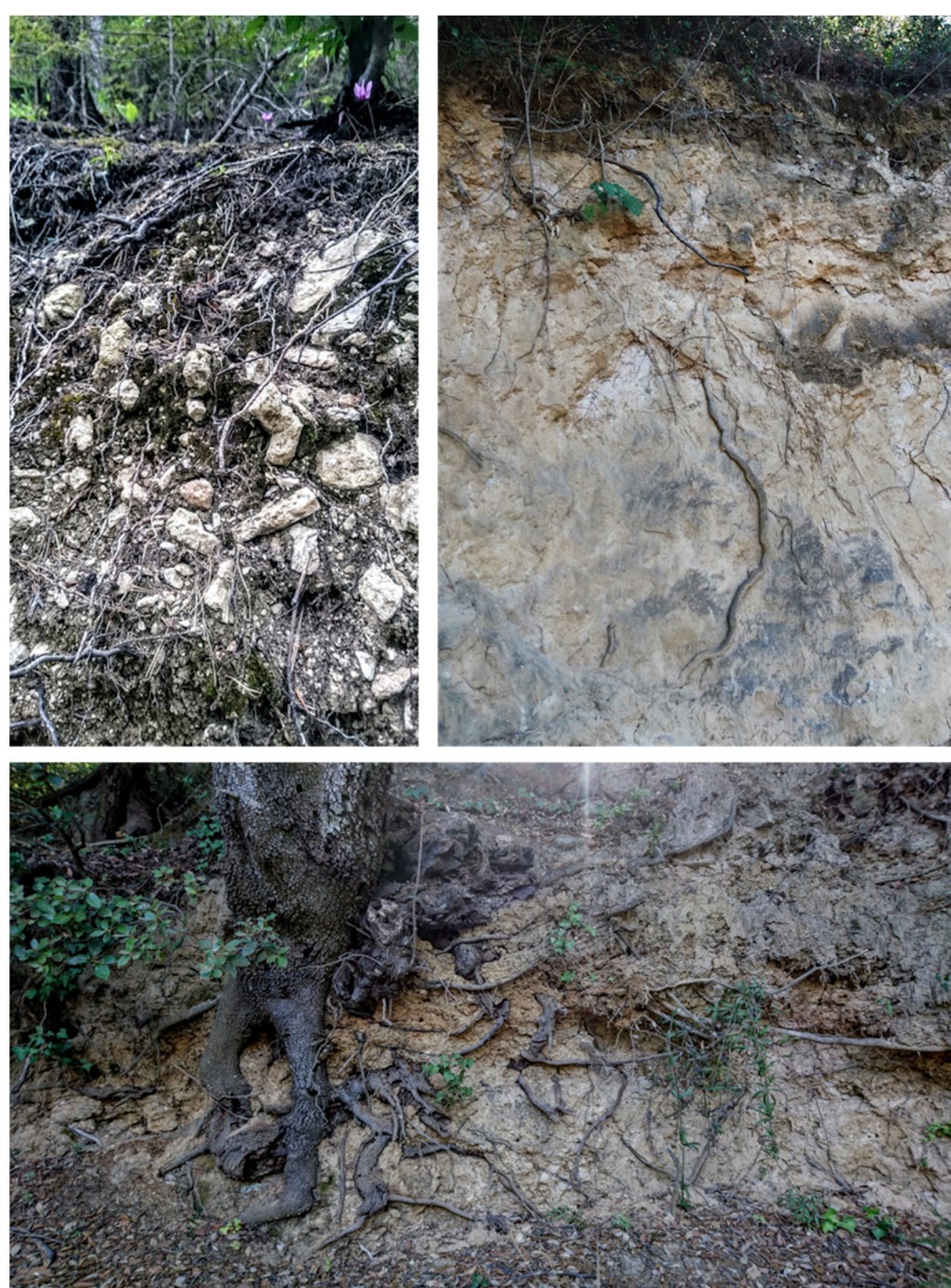

**Figure 2.** Interactions of small and large roots with soil particles and rocks that generate the root reinforcement effect.

### 2.1. Root Reinforcement Effect

Soil reinforcement by roots is recognized as an important factor involved in the forces system that acts on natural and artificial slopes (e.g., [35,36]). The root system represents the part of plants tasked with anchoring (among other fundamental functions) the vegetation to the soil. The combination of earth, roots and bonds forms a reinforced soil matrix, in which stresses can be transferred from the soil to the roots, increasing the overall strength of the matrix [37]. Therefore, the strength of rooted soil depends on soil strength, root strength, and strength of the bonds between soil and roots [23,38,39]. The strengthening effect of a matrix by fibers of different material is achieved only if the two materials have

different tensile and compressive strength properties [40]. In the case of the reinforced matrix soil-roots, the soil is strong in compression, but weak in tension, plant roots are instead weak in compression but strong in tension. The magnitude of root reinforcement is a function of the following factors: (i) root density; (ii) root tensile strength; (iii) root tensile modulus; (iv) root length/diameter ratio; v) soil–root bond strength; (vi) alignment–angularity/straightness of the roots, and vii) orientation of the roots relative to the direction of principal strains (e.g., [41–47]).

Research on the root reinforcement started treating propaedeutic aspects (sustaining the present research lines) represented by exploration of methods to observe roots systems, measure roots strength and distributions, and model rooted soil behavior. Thanks to the pioneering works that have been conducted and the ones that followed until now, mechanical behavior of roots was determined, as well as order of magnitude of common tensile root strength and main control factors of the latter.

Several methods have been individuated to evaluate the mechanical properties of roots. A complete description is beyond the scope of this work and can be found in a recent exhaustive review work by Giadrossich et al. (2017) [48], which also suggests some protocols for laboratory and in situ tests. Typical tensile strength values of roots range from 4 to 20 MPa for grass roots and 5–70 MPa for tree roots [31]. Recently, Abdi (2018) [49] characterized the root tensile force and resistance of eight tree and two shrub species typical of Iranian forests: using two different algorithms, they obtained the power regressions for diameter–force and diameter–resistance relationships of the species under consideration.

Experiments have shown that root tensile strengths decrease exponentially with increasing root diameter (e.g., [12,23,31,34,50–54]). The relationship between root diameter and root tensile strength is usually described using a power-law function of the form:

$$T_r = ad^b, \tag{1}$$

where $T_r$ is the ultimate tensile strength (MPa), $d$ the root diameter (mm), and $a$ and $b$ regression parameters. The power relationship between tensile strength and root diameter can be explained by the scaling effect typical of the fracture mechanics and by another effect suggested by Genet et al. (2005) [50]: an additional explanation can be the different cellulose content in roots of varying diameters. At the smallest root diameters, grass roots have the highest tensile strength and shrubs the lowest. The tensile strengths of grass and tree species tend to converge at root diameters above 5 mm [55]. For root diameters greater than 1 mm, however, grass roots are weaker per unit area than corresponding tree and shrub species. As root reinforcement is also a function of the density of roots in soils, grasses may provide significant reinforcement to the shallower layers of soil where thousands of fine grass roots are concentrated, providing significant reinforcement when potential failure planes are shallow. Conversely, the woody roots of trees and shrubs will provide reinforcement over a greater depth of soil through a combination of both fine, fibrous roots and coarser, woody roots. Different root diameters have a different influence on soil strengthening. During soil shearing, fine roots tend to break, staying in the same position relative to the soil particles. Conversely, coarse roots can be pulled out of the soil without breaking down [39]. A combination of dense fine roots in the top layer (where resistance in tension is important) with coarse, deeply penetrating roots crossing potential shear surfaces is the most efficient configuration to stabilize slopes and riverbanks [56].

Hales et al. (2009) [54] described a possible relation between root tensile strength and hillslope topography. Considering plants belonging to the same species, they found stronger tensile strength in plants located on convex noses of a hillslope relates to the ones located in concave hollows. The hypothesized explanation is the difference in the soil water potential of the topographic locations that is reflected in different root cellulose contents. Although a power-law relation between root diameter and tensile strength can commonly be seen for a given species at a given site, cellulose content, and thus root tensile strength, may also vary with environmental factors. These environmental factors include, but are not

limited to, soil fertility, nutrient supply, soil moisture content, and soil mechanical factors such as bulk.

The age of the plants is another factor affecting the root reinforcement: Dazio et al. (2018) [57] focused on chestnut trees and demonstrated over-aging having a negative affection on the root reinforcement, while coppicing of plants fosters a complete renewal of the root system, with positive effects on slope stability. Based on their results, they concluded that a different management of young chestnut trees could be negligible in terms of root reinforcement, while the difference between young and overaged coppices could be relevant.

The first and most used approach to describe the reinforcement due to roots involved the use of perpendicular root models [24,38] and the integration of the root reinforcement as an additional term in the Mohr–Coulomb shear strength criterion for unsaturated soils [58]:

$$S = c' + (\mu_a - \mu_w) \tan \varphi^b + (\sigma - \mu_a) \tan \varphi\prime + c_r \tag{2}$$

where $S$ is the soil-shearing resistance (kPa), $c'$ the effective cohesion (kPa), $\mu_a$ the pore-air pressure (kPa), $\mu_w$ the porewater pressure (kPa), $\varphi^b$ the angle describing the increase in shear strength due to an increase in matric suction $(\mu_a - \mu_w)$ (°), $\sigma$ the normal stress on the shear plane (kPa), $\varphi\prime$ the effective soil friction angle (°), and $c_r$ the increase in shear strength due to roots (kPa).

Waldron (1977) [38] model assumptions are the vertical extension of all the roots across the horizontal shear zone and the action by the roots as loaded piles (as the soil is sheared, tension is transferred to them). In this model, the tension developed in each root is resolved into a tangential component (that increases the apparent cohesion) and a normal component (that increases the frictional resistance). The angle of each root in relation to the direction of the applied force is, however, important, as this determines the distribution of stresses within the root, and consequently the maximum tensile strength before breaking [59]. Waldron (1977) [38] model is therefore generalized to the case where roots may be oriented at any angle relative to the failure plane [60]:

$$c_r = T_r(RAR)[\sin(90 - \psi) + \cos(90 - \psi) \tan \sigma\prime] \tag{3}$$

where $T_r$ is the root failure strength (tensile, frictional, or compressive) of roots per unit area of soil (kPa), $RAR$ the Root Area Ratio (dimensionless), and $\psi$ the angle of the root at rupture relative to the failure plane (°).

The angle of the root at rupture relative to the failure plane $\psi$ is equal to:

$$\psi = \tan^{-1}\left(\frac{1}{\tan\theta + \frac{1}{\tan i}}\right) \tag{4}$$

where $\theta$ is the angle of shear distortion (°), and $i$ is the initial root orientation relative to the failure plane (°). When root orientation is 90°, the Equation (3) is identical to the model of Waldron (1977) [38]. The use of simple perpendicular root models in cases where it may be assumed that the roots are randomly oriented in the soil is supported by Gray and Ohashi (1983) [61]: they showed that perpendicularly oriented fibers or randomly oriented fibers provided comparable reinforcement. Assuming all roots perpendicular to the failure plane, Wu et al. (1979) [24] selected a constant value of 1.2 to replace the bracketed term (root orientation factor, $R_f$), and Equation (3) became

$$C_r = 1.2\, T_r(RAR) \tag{5}$$

Many researchers have explored the variability of $R_f$ using different assumptions for $\sigma\prime$ and $\theta$: values selected by most authors have tended to be within the range of 1.0–1.2. Based on their studies, Pollen-Bankhead (2010) [7] found that 1.0 is more appropriate than 1.2.

This theoretically based model allows rooting strength to be estimated based on the proportion of the soil area occupied by roots and measurements of the tensile strength of the roots themselves. The model by Wu et al. (1979) [24] tends to overestimate root reinforcement due to the assumptions of full tensile strength of each root mobilized during the soil shearing and the simultaneous breaking of all roots [23,37,52,62,63]. To solve this overestimation, Pollen and Simon (2005) [51] and Pollen (2007) [64] proposed the Fiber-Bundle Model (FBM) (Rip-Root) to consider the progressive root breaking during shear failure. This model used the measured diameters and tensile strengths of roots crossing the shear plane and the constant Rf of 1.2 used by Wu et al. (1979) [24]. The root reinforcement estimated using RipRoot (Cr) was then substituted into Equation (2). Pollen and Simon (2005) [51] and Docker and Hubble (2008) [60] observed that the magnitude of overestimation by Equation (5) was species-specific: the simplified model of Wu et al. (1979) [24] tended to provide better predictions for grasses. Schwarz et al. (2010) [13] applied a FBM approach to modelling the root reinforcement on slope stability in Tuscany, Italy: they confirmed that the Wu model overestimated root reinforcement and, therefore, over-predicted slope factor of safety by up to 10%, with this error increasing exponentially for smaller landslides.

### 2.2. Issues in Considering Root Reinforcement in Slope Stability Models

Nowadays a significant limit persists in properly including the root reinforcement effect into slope stability models, consisting in the difficulties to assess the spatial variations of the root density in soils. Several methods to quantitatively study the roots systems have been experimented and used, but they are highly time-consuming, generally highly invasive or even destructive, and, concerning the most advanced techniques, extremely expensive.

Commonly used methods to determine root density are the excavation of roots systems, the trench profile wall method and techniques based on samples washing for the separation from soils of roots and the detection of the latter [65]. The extraction of soil samples and the successive separation of roots through washing and sieving (the most used method) determines the loss of considerable amounts of roots during its phases [66]. In the last decades, new techniques have been developed, such as automated imaging analyses [67,68], portable minirhizotrons (transparent pipes slantwise inserted in soils, within which video–photo cameras are dropped), color scanner systems [69], and some indirect methods have been searched [70]. However, some techniques are still expensive, quite difficult to be managed, and others have to be developed further.

A main issue in considering the root reinforcement in the slope stability modelling of large areas is the difficulty of evaluating the spatial variations of the parameter, the roots being a complex underground system. The below and above-ground conditions that determine the soil reinforcement are extremely variable in time and space, and the involved elements are countless. Field measurements of root properties in a single hillslope can return results varying of an order of magnitude, even in cases of plants of the same species, sizing, and age [43,45,54,71]. The extreme variability is due to different root biomass and diameter distributions reflecting different distributions of nutritive elements and water, the presence and positioning of eventual physical barriers within the soil, and the different exposure to the sun of the plants [72,73]. The high spatial and temporal variability, the difficulties in the measurements in addition to the uncertainty of models themselves adopted to reproduce the root reinforcement make the consideration of this parameter in the slope stability modelling a very big challenge. Consequently, the international literature is rich in works exploring these research questions. In the next section, we summarize recent advances in estimation of root reinforcement for distributed applications and slope and regional scale modelling considering the root reinforcement.

## 3. Root Reinforcement Modelling: Recent Applications

Literature has been widely enriched lately by results of research on the effects of vegetation on slope stability. The abundant research carried out in the field arose from

different purposes: many works were performed with the finality of developing slope stability models and improving their capability of represent the soil behavior, while for many others, the priority was deepening the knowledge on the vegetation effects for bioengineering purposes. All those studies have in common the consequences of having confirmed, deepened, and expanded our knowledge on the subject, in some cases exploring some aspects not considered in the past. Some authors focused on certain plant species, other on the influence of the forest management, still others on the effect of the moisture gradient and wildfires, exploiting the numerical modelling and/or the field work.

### 3.1. Approaches for Estimating the Root Reinforcement Distribution at a Regional Scale

The spatial distribution uncertainty of root reinforcement that limits regional slope stability models is a well-known problem affecting other physical parameters such as cohesion, friction angle, and hydraulic permeability, to mention a few. It is worthwhile mentioning here that in steep terrain with colluvial soils (very prone to shallow landslides especially at the middle latitudes), friction angles commonly occupy a relatively narrow range of values (few grades), while as aforementioned, root cohesion can vary by an order of magnitude.

Several approaches have been developed and experimented to overcome these limits in considering the root reinforcement at a regional scale. Some of the most recent studies in this field tried to exploit the potentialities of remote sensing techniques to evaluate the root reinforcement through analysis of the aboveground parts of plants. Remote sensing of aboveground parts of plants (geometry, biomass density and health state via vegetation indexes VIs) is advanced enough to result useful to the cause. Indeed, estimation of biomass by remote sensing techniques can be considered a consolidated approach as it can be traced back to the 1970s [74].

A class of remote sensing approaches to the problem based on the VIs which reflect the spectral characteristics of the vegetation–soil system. In recent decades, VIs have been widely used to estimate aboveground biomass in large areas [75–77]. Most common critics moved to this approach to deal with the influence of the background soil on the estimated VIs values. These critics represent the research question of the work by Wang et al. (2019) [78]: their aim was to improve grass-land aboveground biomass assessment modifying 'traditional' VIs—Difference Vegetation Index (DVI), Modified Soil-Adjusted Vegetation Index (MSAVI), Normalized Differential Vegetation Index (NDVI), the Ratio Vegetation Index (RVI)—to minimize the influence of soil background. They estimated the vegetation cover of 156 sites (1 m × 1 m sample plots) modifying the four mentioned VIs (obtaining modified vegetation indexes MVIs) to maximize the differences between vegetated and non-vegetated areas. The MVIs were then regressed with the sample-scale aboveground biomass (AGB) using different functions. They have found that the MVI-AGB models estimate better the AGB than the VI-AGB models and have individuated in the logarithmic MNDVI-AGB model the best one for their study area. Another application of the NDVI to estimate the spatial pattern of root cohesion is described by Chiang and Chang (2011) [79]. In their study on the potential impact of climate change on typhoon-triggered landslides, they used root cohesion values derived from NDVI values for the distributed calculation of the factor of safety. According to the method used by Huang et al. 2006 [80], the authors estimated the spatial variation of root cohesion retrieving the NDVI values and applying a linear transformation to the full spectrum values (−1.0–1.0) by setting the minimum value at 0.0 kPa and the maximum at 50.0 kPa of root cohesion.

A different remote sensing-based approach is represented by the exploitation of the radio electromagnetic waves to estimate the aboveground volumes of plants. Hwang et al. (2015) [81] utilized LIDAR (Light Detection and Ranging) technology to estimate canopy height information and produce a spatially distributed root cohesion model. They developed an approach to characterize spatial patterns of total belowground biomass based on empirically derived allometric relationship developed from soil pit measurements: the vertical distribution of roots and tensile strength (the essential parameters to evaluate the

root cohesion) were sampled at soil pits and related to canopy height. They demonstrated that canopy height information from LIDAR can be effectively used to derive spatial patterns of root cohesion and improve shallow landslides modelling in forested areas.

The inference of the belowground biomass starting from aboveground properties of plants has been also exploited to develop root reinforcement estimate methods not based on the remote sensing. Hales (2018) [82] developed a model to estimate the root reinforcement of slopes, using distributions of biomass measured at the biome level [83], root tensile strength values of different vegetation species from previous studies, and deriving the root densities from the global wood density database [81,84]. The values of root cohesion resulting from the application of the model were subjected to a sensitivity analysis, demonstrating that tensile strength and root density—the parameters determining root cohesion values on slopes—affect the modelled cohesions more than the parameter associated with model uncertainty—the reduction parameter that accounts for the well-known overestimation of root reinforcement by the Wu method. Cislaghi et al. (2017) [85] developed a probabilistic 3D stability model—PRIMULA, PRobabilistIc MUltidimensional shallow Landslide Analysis—that considers the root reinforcement variability following a multi-step procedure: (i) generation of maps with tree locations (a set of random forest configurations for each cell is calculated by Monte Carlo simulations, considering real forest characteristics as density of trees, diameter at breast height, minimal distance between the trees); (ii) calibration of a root distribution model based on field-collected data and the generated tree location maps; (iii) application of a root reinforcement model—the FBM—which combines the density of roots of different diameters within the soil and the mechanical characteristics of roots. The model PRIMULA was subsequently used in the multidimensional approach described in Cislaghi et al. (2018) [86] to quantify large wood recruitment from hillslopes in mountainous catchments. The authors combined forest stand characteristics in a spatially distributed form, the PRIMULA model—the mentioned probabilistic multidimensional slope stability model able to include the root reinforcement— and a hillslope-channel transfer procedure for their purpose of estimate the large wood hillslope-recruitment. This innovative approach was tested in a small catchment in the Alps and provided good results both in terms of identification of unstable areas and large wood volume produced and transferred to the hydraulic channel network.

Arnone et al. (2016) [87] chose instead an eco-hydrological approach suitable for wide areas: they derived the root reinforcement values from eco-hydrologically based estimates of root biomass using a topological root branching model. To estimate the amount of roots and the distribution of diameters with depth (in order to apply the FBM), the authors used Leonardo's rule, according to which the cross-sectional area of a trunk or branch of a tree is equal to the sum of the cross-sectional areas of the branches at any higher level [88]. The approach to the problem adopted by Salvatici et al. (2018) [89] exploited the statistical method of Monte Carlo. The authors performed a distributed and dynamical—considering rainfall intensity data changes during an 8-year long period—slope stability simulation of an 800 km$^2$ wide area in their work aiming to the forecasting of shallow landslides occurrence at a regional scale. The physically based model they applied (HIRESSS, HIgh REsolution Slope Stability Simulator, [90]) has been modified to also consider the root reinforcement provided to soil by the vegetation cover of the study area. The approach to estimate the root reinforcement distribution in such a wide area adopted by the authors was the following: distribution of plant species was first obtained by in situ observation and preexisting vegetational maps, then a value of root cohesion and a range of variation was defined for each plant species based on the most recent literature, finally, the root reinforcement was treated as variable in Monte Carlo simulations (as well as the other geotechnical parameters) to stochastically reproduce its natural variability. Same approach was used also by Cuomo et al. 2020 [91]. In this case, the authors also consider another process works by vegetation in addition to the root reinforcement, they indeed account for the effect of the evapotranspiration on the soil moisture budget in their distributed analyses for shallow rainfall-triggered landslides.

### 3.2. New Slope Stability Models Including Root Reinforcement

In the last decades, research on the role of vegetation in the slope stability has highlighted the importance of the mechanical action of the roots in the mechanism of shallow landslides and many advances have been accomplished to model the effect of the root reinforcement [13,51]. The following step was the introduction of this vegetational effect into stability models and it has been achieved gradually, starting from slope scale applications and achieving successful tests at the regional scale only recently. In the majority of the cases, this aim was persecuted starting from well-developed and largely tested slope stability models such as model TRIGRS (Transient Rainfall Infiltration and Grid–Based Regional Slope-Stability Analysis) [92,93] or HIRESSS [89,91].

In the past, quantitative evaluations of regional slope failure considering contribution of plant cover were very limited [94] because of the lack of purposely developed distributed slope stability models and the limited power of available computers. Recently, high power computers have become a more accessible resource, and research can deal with the development of regional slope stability models considering the root reinforcement. Saadatkhah et al. 2016 [93] consider both hydrological and mechanical effects of vegetation on slope stability of a 25 km$^2$ area through an improved version of model TRIGRS. In their study of a distributed application in a Malaysian region (Kelantan River basin), the authors modified the popular model to also account for rainfall interception by canopy (based on a leaf area index approach), root cohesion and tree weight surcharge. Most part of the parameters used in the analyses were retrieved from literature data. Basing on the results they obtained, the improved version of the model more accurately describes the actual stability conditions of slopes under consideration. During the research conducted by Salvatici et al. [89], the parallel code-based Hiresss model [90] was modified to account for the root reinforcement: this factor was inserted in the geotechnical model as an additional cohesion in the equations of the factor of safety for saturated and unsaturated conditions of soil.

Regarding the slope scale instead, Switala and Wu, 2018 [95] developed a numerical model to perform simulations of rainfall-induced landslides of vegetated slopes, taking into account not only the root reinforcement, but also the evapotranspiration. The numerical model (based on the modified Cam-clay model by Tamagnini, 2004 [96]) was implemented in the finite element codes Comes-Geo for 2D and 3D simulations. The performance of the model is demonstrated by two numerical simulations—2D and 3D—whose results showed the significance of considering the additional strength offered by the presence of vegetation. Successively [97], the authors modelled the stability of a vegetated slope considering both evapotranspiration and root reinforcement using the modified Comes-geo finite element code to execute rainfall triggered landslide simulations. The same slope in two different conditions of vegetation cover—vegetated and not vegetated—was considered, and a constant rainfall intensity to the top of the slope was applied. The results of the numerical modelling confirmed and quantified the effect by mechanical root reinforcement and evapotranspiration in producing slope stabilization and time delay of eventual failures. In their work, they also suggested the appropriate plants distribution on the slope to assure an effective bioengineering stabilization. Cohen and Schwarz (2017) [98] proposed a new slope stability model that considers the effects of tension, compression, and shear in the soil–root system. They tested the model in a synthetic hillslope, finding more complex responses and deformation patterns than those obtained with traditional slope-uniform apparent-cohesion approaches.

Gonzalez-Oullari and Mickowski (2017) [99] dealt with the limits of the available slope stability models (that included the vegetation effects) consisting in the consideration of a few traits of the plants at the time. The authors developed a plant selection multi-module tool for evaluating the capability of different plant species to provide mechanical and hydrological reinforcement to soil in the context of rainfall triggered shallow landslides. The tool permits to combine many different measurable plant traits with varying soil and climate parameters to evaluate plant species influence on slope stability in variable conditions.

### 3.3. Influence of Particular Plant Species on Slope Stability

The variability of the root reinforcement between different species or within the same species between plants of different location and age is huge as already been stated previously in the paper, so that the need to collect quantitative data and develop reliable models to assess the effects of particular plant species is far to be definitely solved. Recent literature is rich in research on selected plant species, and authors have been working in that direction mainly by means of numerical modelling preceded and supported by field measurements and laboratory tests [100–107].

Many recent works have been focusing on vineyards, as they represent a high-value plantation that is often cultivated in steep hills and whose effect on slope stability has not been particularly explored. Bordoni et al. (2016) [100], Cislaghi et al. (2017) [101], and Bordoni et al. (2020) [102] explored the characteristics of the root reinforcement provided by this plantation. Bordoni et al. (2016) [100] carried out a study aimed to quantitatively evaluate the soil reinforcement provided by the vineyards in a study area (13.4 km$^2$) characterized by high shallow landslides density. The authors evaluated root density and root mechanical properties in seven test sites to estimate the root reinforcement distribution, in addition to perform a geotechnical characterization of the soils. They were able to observe root tensile strength measures only slightly variable in space and time, root densities having a marked variability in the soil profile, and lower densities of roots in sites characterized by less permeable soils (which are more often associated to landslides in the study area). Cislaghi et al. (2017) [101], interested in the role of vineyards in the slope stability, developed a model to evaluate the soil reinforcement due to grapevine roots. They modelled the additional reinforcement (and its spatial distribution) provided by the grapevines adopting the FBM and a root distribution model. They calibrated the two models for their vineyards by means of in situ measurements and laboratory tests. The estimated root reinforcement distribution was then considered to perform a series of back-analyses on some well documented landslides of an Italian site using a limit equilibrium stability model. They found that the spatial variability of root reinforcement was negligible for this kind of plantation (due to the extreme regularity of vineyard plantations) and that some features of the plantation design namely plant density, row distance, and rootstock depth could be optimized to provide a positive support to slope stability. More recently, Bordoni et al. (2020) [102] reported a study in which root properties—root density, mechanical properties, and reinforcement—of different land use types—sowed grasses used to produce animal feed, vineyards cultivated with different agronomical management practices, shrublands, woodlands of broadleaved species—were evaluated in 38 test sites to analyze the influence of different land uses on shallow landslides probability. The root reinforcement estimates were used to perform punctual deterministic (first) and probabilistic (then) slope stability analyses in which two soil type characterized by different geotechnical properties were considered. The authors found lower failure probabilities for the land uses "vineyards with permanent grass cover", "vineyards with alternation in the inter-rows", and "woodlands", highlighting the importance of land use managements for shallow landslides mitigation purposes.

Wang et al. (2019) [103] examined the effects on slope stability of three tree species— *Populus tomentosa*, *Robinia pseudoacacia*, *Olea europaea*—in a geohazard-prone region in China. They assessed parameters such as root density, variation of RAR with depth, root tensile strength, root peak tensile force, and the contribution of the root cohesion to the factor of safety. They used Wu–Waldron and fiber bundle approaches to model the root cohesion, and a 2D finite element model for the stability analysis in order to investigate the contribution of the root cohesion to the factor of safety. They pointed out that the mechanical reinforcement of root systems exhibited a vertical variation and introduced a depth-dependent RAR in a slope-scale stability model to investigate the effect of various species on hillslope stability.

Chiaradia et al. (2016) [104] analyzed the contribution of three widely European species—European beech, sweet chestnut, Norway spruce—to the slope stability by means

of field measurements on roots and statistical tests. They performed root distribution analyses in 15 different test sites (5 for each species), at each station root tensile forces were evaluated on samples of the three species. They then evaluated the contribution of the species to the slope stability performing simulations with a probabilistic limit equilibrium slope stability model. They found that the best approach to input the root reinforcement in the probabilistic framework was a lognormal function in all three cases.

Wang et al. (2017) [105] explored the possibility to use vegetation as an alternative countermeasure to reduce debris flows hazard. Focusing on *Robinia pseudoacacia*, they performed field investigations and statistical analyses to characterize the root system, then applied this characterization to a Bank Stability and Toe Erosion Model and to a cellular braided-stream module, demonstrating that the tensile strength of the roots increases with time, leading to an increased stability of the hillslopes, a stabilization of the morphology of the channels, and a reduction of the extension of the areas hit by debris flows.

Likitlersuang et al. (2017) [106] focused on the effect of vetiver grass, a plant species that became popular in bioengineering for hillslope reinforcement purposes. They combined numerical modelling and laboratory tests (using a geotechnical centrifuge and a rainfall simulator) to quantify the beneficial effects of vetiver, consisting in the reduction of infiltration rates, delay in groundwater rise, and increase of soil shear strength.

Rossi et al., 2017 [107] applied the distributed slope stability model LAPSUS _LS (LandscApe ProcesS modelling at mUlti dimensions and scaleS [108]) in a small catchment in Costa Rica and explored the sensitivity of the results to various input parameters, including root reinforcement and biomass surcharge provided by coffee plantations (and coffee plantation mixed with deep-root vegetation). They found the biomass surcharge was almost uninfluential, while the root reinforcement has proven a significant parameter affecting the results, especially if roots were deeper than the shear plane.

Gonzalez-Oullari and Mickowski (2017) [99] used their Plant-Best model to assess the hydrological and mechanical soil reinforcement of seven plant species (*Acer pseudoplatanus* L., *Fraxinus excelsior* L., *Salix* sp., *Fagus sylvatica* L., *Quercus* sp., *Silene dioica* Clariv., *Erigeron acris* L.) in different morphological, geotechnical, and climate conditions. They could evaluate the points of force of each species from the slope stabilization viewpoint, revealing, amongst others, that different plant species were suitable for protection depending on the hydrological conditions and, generally speaking, underweight plants with dense root systems and broad thick canopies are preferable for bio-engineering stabilization purposes.

### 3.4. Influence Forest Structure, Wildfires, and Soil Moisture Gradient

Despite all advances made since the research on vegetation impact on slope stability has started, the variables implicated are so many that aspects that are even potentially very significative have been considered or deepened only recently. Relevant aspects that have been deepened only recently include, e.g. how the relationships between root bundles and soil moisture conditions affect slope stability, or the influence of stand structure changes due to human/natural disturbances like silviculture, seasonal variations, and wildfires on the protective function of forests have been deepened recently.

Forest structural characteristics such as extent and size of forest gaps, ages of stand or species composition are known to be influent on slope stability [109,110]: being dynamical factors subjected to natural and human-driven disturbances, the quantification of their effect on landslide susceptibility is an essential research aim [3,13,93]. Moos et al., (2016) [111] investigated the influence of the forest structure on slope stability applying statistical prediction models and a physically based model to estimate distribution of root reinforcement. The authors could find that forest structure variables (in addiction to geohydrological characteristics of terrains) significantly influence landslide susceptibility. The most significant forest structure variables resulted: gap length—breaks in the forest canopy, particularly critical for landslide initiation if higher than 20m; distance to the nearest tree; a proxy-variable for root reinforcement of the nearest tree. Rickli et al. (2019) [112] investigated the forest structure influences by means of a back-analysis approach instead,

analyzing the characteristics of more than 600 landslides and their vegetation cover of the Swiss territory. The analysis of the data they collected confirms that the presence of forest on the slope can be reasonably considered as a key factor in terms of stability, especially in steep slopes—more than 35° of gradient—and underlines the importance of the forest structure characteristics such as spacing between trees and dimension of gaps among vegetation. Schmaltz and Mergili (2018) [113] conducted a sensitivity analysis of the slope stability to different forest stand configurations and density of root systems. Using a GIS-based slip surface model, they tested 23 different root system scenarios and a generic hillslope landscape combining configurations of roots—shallow, taproot, mixed—with different stand and patch densities, and considering different size of potential shallow slip surfaces. The sensitivity analysis showed that the stand density, the approximate position on the slip surface and the ability of the root system to cross the potential slip surface were consistent driving factors for the stability model outputs.

Other authors studied the influence of the forest structure on the landslide susceptibility focusing specifically on the effects of human actions related to forest management and harvesting. Cislaghi et al. (2019) [114] assessed the effects of different forest management strategies on the slope stability considering the root reinforcement differences. They studied a peculiar distribution function for the root reinforcement according to the forest stands characteristics and the forest management to analyze the slope stability of a 1.1 km$^2$ wide area with a probabilistic physically based model. They found that thinning procedures can have a significant impact on slope stability, with crown thinning causing a significant reduction of root reinforcement and low and intermediate thinning causing only small decreases, independently from intensity of thinning. Vergani et al. (2016) [115] focused on the spatial and temporal variability of root reinforcement after forest harvesting instead. They estimated the root reinforcement by means of field testing on roots and the application of the Root Bundle Model in order to develop a root reinforcement decay model caused by harvesting procedures. They discovered that the root reinforcement provided by spruce trees decayed by 60% in 5 years and completely disappeared after 15 years, partially compensated by regrowth and shrubs that guarantee 30% of the original reinforcement.

The natural seasonally variability in the foliage was also found to be a not negligible factor affecting the root reinforcement, as Schmaltz et al. (2019) [116] suggested with their study. The authors have studied the importance of considering the land cover dynamics in the slope stability analyses, and in the application of a slope stability model, different approaches with growing complexity were used to describe the vegetation. They could find that completely neglecting the effect of vegetation as predictable has led to an overestimation of instability; conversely, the introduction of widely used simple modules accounting for a generalized effect of trees has led to an underestimation of instability. The best results were obtained using dynamic models that account also for the development stages and the seasonal variations of the main vegetated land cover classes.

The possible change of roots strength with soil moisture variations is reasonably hypothetical, but to date, there are few studies on the argument that can support the researchers in considering that aspect in the slope stability analyses. A recent study on the effect of soil moisture gradient on the roots' strength was carried out by Hales and Miniat (2017) [43]. The authors demonstrated a strong dependence of spatio-temporal variations in root reinforcement from soil moisture gradients. They performed laboratory and field test in hillslopes of different topography and hydrologic characteristics—i.e., convergent hollows and divergent noses—finding that an increase of soil moisture led to a reduction of root tensile strength (and thus of root reinforcement) and a consequent increase of hillslope instability. They suggested that the tensile strength of root systems dynamically changes during a rainstorm, resulting in relevant reduction of the factor of safety. Wang et al. (2018) [117] were also interested in the influence of moisture on the root reinforcement, but approached the problem by means of laboratory and numerical experiments. They investigated the changes in the interfacial friction between soil and

grass roots during precipitation, finding that grass can delay the time of occurrence of landslide initiation, but that effect diminishes as the soil moisture increases.

The root reinforcement is likely to be affected also by wildfires: plant damages and deaths caused by wildfires imply progressive root decomposition processes, resulting in a decrease or total disappearance of their stabilizing action [118,119]. Although wildfire is a frequent and intense disturbance factor for forests throughout the world [120,121], quantitative studies on the impact on the root reinforcement and its recovery are still scarce [121]. Recently, Vergani et al., 2017 [118], interested in the topic, quantified the temporal and spatial dynamics of root reinforcement in a Scots pine (*Pinus silvestris*) four years after a stand replacing fire. They compared roots distribution measurements and pull-out tests of the burnt stand with those performed in a nearby healthy stand of the same age. They demonstrated that four years after the replacing fire, the protective capacity of the stand was highly compromised, and regeneration was not able to counterbalance the root reinforcement losses. The consequences of wildfire on the root reinforcement in time was also evaluated by Gehring et al., 2019 [121] at multiple timesteps in this case. They evaluated magnitude and time-variability of the root reinforcement decrease caused by wildfires, finding a root reinforcement decrease that varied greatly according to fire severity. Basing on their findings, in areas severely hit by wildfires, the protective capacity due to root reinforcements decreases rapidly, reaching zero within about 15 years after the wildfires, and may remain zero for at least two decades.

## 4. Discussion

An overview of the studies described in the previous section reveals that including the mechanical effect of root systems into slope stability analysis is a research topic that is receiving growing attention. Table 1 provides a summary of the revised studies published during the years 2015–2020 regarding the root reinforcement (mainly treated in Section 3). Papers have been classified based on adopted approaches and scale of models to have an overall view of the orientation of recent research in the field (classes are not mutually exclusive). The category "Roots Modeling" includes works in which the modelling of root systems represents the main objective of the research or, even if it is one of the main objectives, a large part of the research efforts was spent for that, creating models in scale of plants or root systems, or collecting large amounts of in situ information, and deeply analyzing data in order to develop new and advanced models to describe the roots behavior in the context of slope stability analyses. Being a summary of the research on the root reinforcement, the works of the classes "Slope or Smaller Scale SSM (Slope Stability Modeling)" and "Basin or Larger Scale" not classified as "Roots Modeling" also obviously have gone through a certain grade of root reinforcement modeling, but in these cases, the root reinforcement did not represent the focus of the research.

**Table 1.** Content of the papers dealing with the root reinforcement (2015–2020). "Measures on roots" refers to measurements carried out on plant roots both in field or laboratory; "Roots Modeling" represents thorough modeling of roots systems and their behavior; "Slope or Smaller Scale SSM (Slope Stability Modeling)" refers to slope stability analyses carried out by means of models of slopes—numerical modelling of slopes or slopes reproductions in laboratory ad centrifuge tests; "Basin or Larger Scale SSM" refers to slope stability analyses about basins or larger areas; "Other" is for statistical back analyses and reviews.

| Authors | Measures on Roots | Roots Modeling | Slope or Smaller Scale SSM | Basin or Larger Scale SSM | Other |
|---|---|---|---|---|---|
| Abdi et al. 2018 [49] | x | | | | |
| Arnone et al. 2016 [87] | | x | x | | |
| Bordoni et al. 2020 [102] | x | x | | x | |
| Bordoni et al. 2016 [100] | x | x | | | |
| Chiaradia et al. 2016 [104] | x | x | x | | |
| Chok et al. 2015 [36] | | x | x | | |
| Cislaghi et al. 2017 [101] | x | x | | | |
| Cislaghi et al. 2017 [85] | x | x | | x | |
| Cislaghi et al. 2018 [86] | | | | x | |
| Cislaghi et al. 2019 [114] | x | x | | x | |
| Cuomo et al. 2020 [91] | | | | x | |
| Dazio et al. 2018 [57] | x | x | | | |
| Gehring et al. 2019 [121] | x | x | | | |
| Giadrossich et al. 2017 [48] | x | | | | |
| Gonzalez-Ollauri 2017 [99] | x | x | | x | |
| Hales et al. 2018 [82] | | x | | x | |
| Hales and Miniat 2017 [43] | x | x | x | | |
| Hwang et al. 2015 [81] | x | x | | x | |
| Kokutse et al. 2016 [35] | | x | x | | |
| Likitlersuang et al. 2017 [106] | | | x | | |
| Masi et al. 2020 [70] | x | | | | |
| Moos et al. 2016 [111] | x | x | x | | |
| Rickli et al. 2019 [112] | | | | | x |
| Rossi et al. 2017 [107] | | | | x | |
| Saadatkhah et al. 2016 [93] | | x | | x | |
| Salvatici et al. 2018 [89] | | | | x | |
| Schmaltz and Mergili 2018 [113] | | x | x | | |
| Schmaltz et al. 2019 [116] | | x | | x | |

**Table 1.** *Cont.*

| Authors | Measures on Roots | Roots Modeling | Slope or Smaller Scale SSM | Basin or Larger Scale SSM | Other |
|---|---|---|---|---|---|
| Switala and Wu 2018 [95] | | | x | | |
| Switala and Wu 2019 [97] | | | x | | |
| Vergani et al. 2017 [11] | | | | | x |
| Vergani et al. 2017 [118] | x | x | | | |
| Vergani et al. 2015 [21] | x | x | | | |
| Wang et al. 2017 [105] | x | x | x | | |
| Wang et al. 2018 [117] | | x | x | | |
| Wang et al. 2019 [103] | x | x | x | | |
| Total Papers: 36 | Count:  19 | 24 | 13 | 12 | 2 |

In the last few years, a challenging research trend emerged in the field of slope stability modelling: the progressive widening of application areas from slope to catchment and regional scale—i.e., over dozens or hundreds of squared kilometer-wide areas. Although advances made in the modelling of the mechanical behavior of roots in soils [13,51], in the evaluation of root tensile strength [44,47,50], and in the spatial distribution modelling of root reinforcement at the stand level [3,54], we could find that estimating the root reinforcement at the regional scale remains challenging, and it represents one of the main limits of the distributed slope stability analyses in wide areas. Slopes scale methodologies of analysis are quite well established, applications over small catchments have become recurrent, but large—dozens or hundreds of km²—areas are investigated only in a few recent works: Cuomo (2020) [91], Hales (2018) [82], Hwang (2015) [81], Salvatici (2018) [89], Rossi (2017) [107], and Saadtkhah (2016) [93]. Studies reporting on slope scale analyses or numerical simulations of synthetic slopes are also essential for the scientists working on wider areas, as they investigate new parameterization strategies, new modeling approaches and sensitivity issues. The great challenge that comes after consists in transferring efficiently lessons learnt from local scale studies to wider areas of interest for integrated land management purposes or landslide hazard prevention. In this context, measurements of roots characteristics—strength and spatial densities—through direct and indirect tests becoming even more expensive and time consuming because of the larger areal extent and the need to account for a larger spatial and temporal variability, as resulted of larger areas, indeed, heterogeneity of conditions that affect the root characteristics increases.

The huge spatial variability of root reinforcement has been shown to be a common feature of almost all areas and plants investigated by literature. The consequent challenge posed to the research of finding indirect methods to evaluate such parameter in wide areas resulted in the development of various approaches based on remote sensing and inference technics starting from aboveground parts of plants [76–79]. Such approaches allow the application of slope stability modelling in increasingly large areas using the distributed slope stability models recently modified to account for the root cohesion [87,91].

The large recent research that has been dedicated to quantitatively assess the influence on slope stability by particular plant species [100–107] added essential information to our knowledge about the actions exerted by plants of interest for their large distribution in nature or agriculture or their common use in bio-engineering, providing useful information about how to improve approaches to input the root reinforcement in the models or which species are suitable for stabilization depending on the context. Furthermore, the results confirmed the high variability of the mechanical behavior of plants, highlighting the importance of keeping active this research topic by continuing to study new plant species or those already studied, but in different environmental conditions.

Plants are complex organisms highly interacting with atmosphere, hydrosphere, lithosphere, and biosphere so that the aspects interesting for slope stability are countless and even some potentially very significative ones such as the influence of forest structure, wildfires, and soil moisture gradient have been considered or deepened only recently [43,111–121]. The research that dealt with the effects of forest structure variables on slope stability such as gap length or distances between trees [111–115] highlighted that these factors are not negligible in order to contain the landslide hazard and that, therefore, should be taken into account by forest management operators. Despite wildfires is a worldwide common and highly destructive factor of disturbance for vegetation, quantitative data on the impact on root reinforcement and its post-fire evolution are still scarce. In this context, the research carried out by Vergani et al., 2017 [118] and Gehring et al., 2019 [121] that focused specifically to the impact of wildfires on the root reinforcement highlighted how much the root reinforcement is highly and long impacted particularly in case of high severe fire. This effect is particularly important to account for in physically based models of slopes affected by wildfires because it comes in addition to the changes induced on the hillslope hydrological system [122]. Furthermore, the authors could find different responses of the root reinforcement depending on plant species and fire severity, bringing out the impor-

tance of further studies on this topic. Studying the influence of changes in the soil moisture content on the root reinforcement is important for two main reasons: moisture content variations happen on large scale, and rainfall is the main triggering factor for shallow landslides. Therefore, studies on the root reinforcement behavior responding to moisture gradient are much needed to support the rainfall-triggered landslides modelling. The root reinforcement is commonly modelled as a component of the soil cohesion in the slope stability analyses [36,43,91,93,123], so that during the stability simulations it decreases with increasing soil moisture following the same relations adopted for the other components of soil cohesion. Considering that the phenomena affecting the root cohesion decreases with soil moisture increases are different from those affecting the soil particles, further studies on this topic in addition to the Hales and Miniat (2017) [43] and Wang et al., (2018) [117] research would be very helpful to properly consider the root reinforcement in the slope stability modelling.

As we have seen in the above discussions, applying recent research outcomes to other case studies would be useful to get a more robust verification and validation of the methodologies of analysis and modeling, and to investigate the possibility to transfer them to wider areal extents and different settings. This last point is particularly important as the modeling of root reinforcement effect into slope stability analyses depends on geological, climatological, and vegetational characteristics of study areas, making highly site-specific the results of any work. Considering this, we felt it beneficial to keep track of countries and application contexts of recent root reinforcement studies, analyzing the geographical location of the reviewed papers (reference period: 2015–2020). Results are portrayed in Figure 3: most of the recent literature is carried out by European institutions with an emerging role of Asiatic research groups. Most part of the reviewed studies are based in the Alps, a geographic macro-region that includes works coming from at least four different countries—Austria, Switzerland, France, and most of the Italian papers. This is not surprising as the area is a well-known hotspot of landslide hazard [124,125] and includes institutions that pay great attention in forest integrated management [126,127].

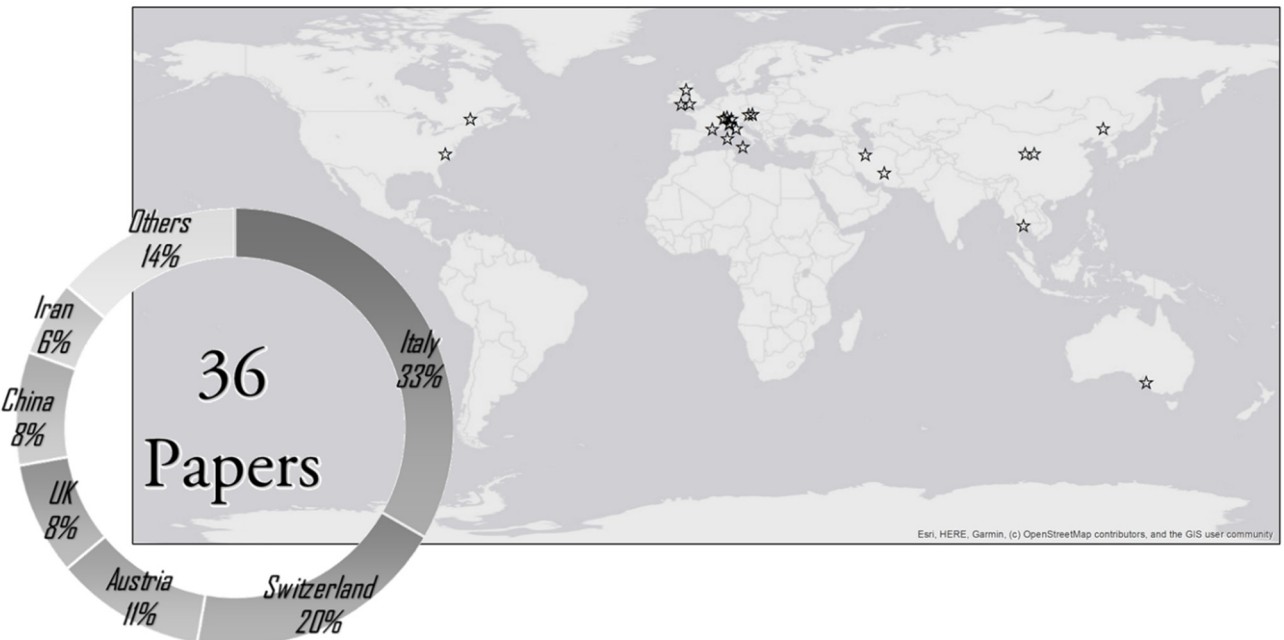

**Figure 3.** Worldwide distribution of the revised studies published during the period 2015–2020 regarding the root reinforcement. Points are located based on corresponding author's affiliation.

## 5. Conclusions

The many studies dealing with root reinforcement that have lately enriched the literature arose from different specific purposes, as improving capability of slope stability model to represent the soil behavior or deepening the knowledge on the vegetation effects for bioengineering purposes, but all contributed to expand the general knowledge on a subject useful to develop increasingly complex and complete models for slope stability analyses. One of the most relevant accomplishments of this review was to bridge together outcomes, practices and methods originated from different subtopics and fostering a cross-fertilization that could be useful to better address future research questions.

Since the vegetation influences have been considered and studied in the slope stability perspective, many aspects interesting the mechanical and hydrological soil behavior have been explored and as many advances have been made. Literature review has shown that the first research topics regarding the root reinforcement to be treated by scientists has been the development of methods to measure roots strength and distribution, being such topics propaedeutic to all the successive. Subsequently, efforts have been dedicated to our ability to model the mechanical root behavior in soils and to model the spatial variability of root reinforcement at the stand level. Reviewing the recent literature on the argument, we could observe that knowledge about the just mentioned topics reached a level for which research trends are now moving mainly in other, but closely related, directions. We found as present main research lines: the research in indirect methods to estimate the root reinforcement by means of remote sensing; the introduction of the root reinforcement into slope stability models (possibly coupled to interception by canopies and evapotranspiration effects) both at slope scale and basin/regional scale; application of models to assess the influences on slope stability by particular plant species; the exploration of almost-new aspects such as impacts of changes in forest structure caused by sylviculture or wildfires and moisture gradient on the magnitude and variations of root reinforcement.

The vast spatial and temporal variability characterizing the root reinforcement still represents an open challenge for research in distributed slope stability modelling of wide-areas and every new research in the field is much needed. The results of the studies conducted to assess the root reinforcement impact of different plant species highlighted the high species-specific character of the parameter. That points out the importance to pursue the study of new plant species root reinforcement impacts as well as already studied plant species, but in different environmental conditions. The impact of forest structure disturbances due to sylviculture or wildfires on root reinforcement emerged as significative and further studies are therefore needed in this direction. Lastly, some recent works pointed out that soil moisture has a significant control on root tensile strength. Considering that moisture content variations happen on large scale and are closely related with the initiation of landslides, further studies on this regard could be highly beneficial for a thorough integration of root reinforcement in slope stability models.

**Author Contributions:** Conceptualization, E.B.M., S.S. and V.T.; methodology, E.B.M. and S.S.; formal analysis, E.B.M.; investigation, E.B.M., S.S. and V.T.; data curation, E.B.M. and V.T.; writing—original draft preparation, E.B.M.; writing—review and editing, E.B.M., S.S. and V.T.; visualization, E.B.M.; supervision, S.S. and V.T. All authors have read and agreed to the published version of the manuscript.

**Funding:** This work was funded by "Dipartimento della Protezione Civile—Presidenza del Consiglio dei Ministri" (Presidency of the Council of Ministers - Department of Civil Protection); this publication, however, does not reflect the position and social policies of the Department.

**Conflicts of Interest:** The authors declare no conflict of interest.

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
