# Peer review of "Root Reinforcement in Slope Stability Models: A Review"

_geosciences, doi:10.3390/geosciences11050212_

Round 1

Reviewer 1 Report

The manuscript reviewed more than 100 articles on root reinforcement; however, it still needs some effort to be published. A good review article should do the hard work of summarizing, critiquing, and synthesizing the research that has been done on a particular topic. These discussions (summarizing, critiquing, and synthesizing) are lacking in the manuscript. Especially, the presentation of section 3 has many defects that need to be improved before the manuscript could become worthy of publication. I hope the authors will try to use some graphs and figures to explain the outcomes of your reference studies. Below, there are some issues the authors must resolve before accepted.

  1. The manuscript shows different types of applications on root reinforcement in section 3. However, these studies should be well compared. I recommend that the authors should explain the differences between the studies. In addition, good scientific writing tells a story, so try to integrate independently described references and make it like a story.

  1. Please elaborate on the latest developments, current research dilemmas, and future trends. They are important for a review article.

  1. The timeline of root reinforcement research should be described more clearly, and the key research in the domain/field should be emphasized.

Author Response

Reviwer#1:

The manuscript reviewed more than 100 articles on root reinforcement; however, it still needs some effort to be published. A good review article should do the hard work of summarizing, critiquing, and synthesizing the research that has been done on a particular topic. These discussions (summarizing, critiquing, and synthesizing) are lacking in the manuscript. Especially, the presentation of section 3 has many defects that need to be improved before the manuscript could become worthy of publication. I hope the authors will try to use some graphs and figures to explain the outcomes of your reference studies.

Below, there are some issues the authors must resolve before accepted.

The manuscript shows different types of applications on root reinforcement in section 3. However, these studies should be well compared. I recommend that the authors should explain the differences between the studies. In addition, good scientific writing tells a story, so try to integrate independently described references and make it like a story.

Authors: Section 3 was profusely revised, references were better integrated and new paragraphs and statements have been added to compose the story suggested by reviewer.

Please elaborate on the latest developments, current research dilemmas, and future trends. They are important for a review article.

Authors: Discussions and conclusions have been expanded a lot deepening current research questions and future trends and are now presented as separated sections.

The timeline of root reinforcement research should be described more clearly, and the key research in the domain/field should be emphasized.

Authors: Section 2.1 was streamlined and better arranged, and a paragraph that clear the beginning of the story of the research on the root reinforcement was added. The complete timeline of root reinforcement has been retraced, discussed and summarized in discussions and conclusions sections. We think that on the whole root reinforcement is now more clearly described and key research emphasized.

Reviewer 2 Report

The paper reviews works dealing with the role of root system in slope stability of soils due to shallow instabilities. Interestingly, the authors examine the contribution of root system in mechanical and hydrological behavior of soils mainly in terms of soil reinforcement. In addition, an abundant of parameters related on root systems (type of plant, root system density etc.) are discussed, as well as different ways to determine them, useful for soil evaluation. Moreover, some interesting examples present the footprint of root system in relation to forest structure, landslide susceptibility or post-wildfire affected areas.

Comment of minor significance

A short task for the effect of roots in rock mass quality of slopes could be added improving the whole range of root system impact in slope stability.

Author Response

Reviewer#2:

The paper reviews works dealing with the role of root system in slope stability of soils due to shallow instabilities. Interestingly, the authors examine the contribution of root system in mechanical and hydrological behavior of soils mainly in terms of soil reinforcement. In addition, an abundant of parameters related on root systems (type of plant, root system density etc.) are discussed, as well as different ways to determine them, useful for soil evaluation. Moreover, some interesting examples present the footprint of root system in relation to forest structure, landslide susceptibility or post-wildfire affected areas.

Comment of minor significance

A short task for the effect of roots in rock mass quality of slopes could be added improving the whole range of root system impact in slope stability.

Authors: We thank reviewer for the suggestion. We agree that the effect of roots on rocky slopes is a very interesting topic, moreover, needing and deserving much attention by the research since to date only few works have been dedicated to it. Anyway, this represents an effect of vegetation that we think fits better with different mechanisms than “root reinforcement” as “anchoring” or “rock weathering” towards whom the review was not directed, but we have added two points in the bullet list of the mechanical effects in Section 2.

Round 2

Reviewer 1 Report

The manuscript has been revised based on the reviewer's suggestions. It is ready to be published now.